# Oxidation of Cyclohexane/Cyclohexanone Mixture with Oxygen as Alternative Method of Adipic Acid Synthesis

**DOI:** 10.3390/ma16010298

**Published:** 2022-12-28

**Authors:** Dawid Lisicki, Beata Orlińska, Adam A. Marek, Jakub Bińczak, Krzysztof Dziuba, Tomasz Martyniuk

**Affiliations:** 1Department of Chemical Organic Technology and Petrochemistry, PhD School, Silesian University of Technology, Akademicka 2A, 44-100 Gliwice, Poland; 2Grupa Azoty Zakłady Azotowe, “Puławy” S.A., Al. Tysiąclecia Państwa Polskiego 13, 24-110 Puławy, Poland

**Keywords:** adipic acid (AA), cyclohexane (CH), cyclohexanone (CH=O), oxidation, *N*-hydroxyphtalimide (NHPI)

## Abstract

Herein, an alternative method for adipic acid (AA) synthesis of industrial importance has been reported. The proposed novel method involves the one-step, solvent-free oxidation of a cyclohexane/cyclohexanone (CH/CH=O) mixture, with a cheap oxidizing agent such as O_2_ or air under mild conditions in the presence of *N*-hydroxyphtalimide (NHPI) and transition metals as catalysts. It has been showed that CH/CH=O mixture under applied mild conditions oxidized faster than CH and CH=O separately. This was due to the greater oxidizability of CH=O compared to CH. The peroxyl radicals formed by CH=O oxidation initiated the oxidation of the less reactive CH. Additionally, CH=O increased the polarity of the reaction mixture, promoting the solubility of NHPI. The influence of type and amount of catalyst, cyclohexane to cyclohexanone ratio, temperature, time, type of oxidizing agent on the composition of CH/CH=O oxidation products have been reported.

## 1. Introduction

Adipic acid (AA), the most important dicarboxylic acid in terms of production volume, is currently generated in approx. 3 million tons per year and is mainly used in nylon 66 production. In industry, AA is produced primarily from cyclohexane (CH) via a two-step process: oxidation of cyclohexane to cyclohexanone/cyclohexanol mixture (K/A), followed by oxidation of cyclohexanone or K/A to AA (Figure 1) [1,2].

CH oxidation is conducted with air in the liquid phase at a temperature of approx. 160 °C and under a pressure of 1–2 MPa in the presence of soluble cobalt(II) and iron(II) salts. The significant disadvantages of this method are the low conversion of the raw material, amounting to 5–10%, in order to ensure high K/A selectivity and separation of significant amounts of cyclohexane. The acquired K/A mixture then undergoes oxidation using 50–60% nitric acid (HNO_3_) as the oxidizing agent and copper(II) and ammonium metavanadate as catalysts at 60–100 °C under a pressure of 0.1–0.4 MPa. Complete conversion of K/A and high AA yield of about 95% can be achieved at this stage. However, the use of HNO_3_ as an oxidant generates significant amounts of nitrogen oxides, including strong greenhouse gas N_2_O at approx. 300 kg per ton of AA. N_2_O cannot be recycled to HNO_3_, hence, it requires implementation of N_2_O abatement technology, which generates additional costs.

Other, much less popular industrial methods use phenol or cyclohexene as raw materials, which, like cyclohexane, are produced from benzene (Figure 1). Cyclohexanol, generated therefrom via hydrogenation or hydration, respectively, is in-turn oxidize to AA again using HNO_3_.

The shortcomings of known industrial methods have led to intensive research into the development of alternative, environmentally friendly, economically viable methods of AA production [3]. Among reported alternative methods that employ petrochemical raw materials, more attention is paid to technologies that use oxygen or air as oxidizing agent. Promising methods with industrial potential include oxidation of cyclohexanone (CH=O) with O_2_ or air and direct oxidation of CH with air to AA. To-date, a wide variety of homogeneous and heterogeneous catalysts of different activity, price and availability have been reported for these oxidation processes.

For example, 99% CH=O conversion and 64% AA selectivity were achieved when CH=O oxidation reaction was carried out using O_2_ in the presence of homogeneous catalyst Mn(acac)_2_/*N*-hydroxyphthalimide (NHPI) with AcOH as the solvent (100 °C, 0.1 MPa, 6 h) [4]. Cavani used Keggin-type polyoxometalates as a catalyst with H_2_O/AcOH (1/1 *v*/*v*) mixture as solvent, producing approx. 65% AA selectivity and 65% CH=O conversion (70 °C, 0.1 O_2_ MPa) [5]. Oxidation of CH=O under an air atmosphere, at 5 MPa pressure and using water as a solvent in the presence of carbon supported platinum catalyst Pt/C at 140 °C led to AA formation in approx. 38% yield [6].

Direct oxidation of CH to AA using O_2_ was also carried out in the presence of Mn(acac)_2_/*N*-hydroxyphthalimide (NHPI) catalytic system using AcOH as a solvent. Authors achieved 36% CH conversion and 46% AA selectivity (100 °C, 0.1 MPa, 6 h) [4]. Recently, Bal at al. obtained AA in 17% yield and 40% CH conversion after oxidation using air in the presence of heterogeneous nanostructured catalyst Co_3_O_4_@ZrO_2_ (130 °C, 1 MPa, 8 h) [7]. Additionally, the combination of Fe-filled carbon nanotubes was used with a small amount of CH=O (2.8% by weight) in a mixture of acetone and butanone as solvent for the direct CH oxidation, which gave 61% AA selectivity and 37% hydrocarbon conversion (125 °C, 1.5 MPa, 8 h) [8]. Furthermore, heterogeneous Ni-SAPO-34 molecular sieve was employed as a catalyst for direct CH oxidation [9].

Hydrogen peroxide (H_2_O_2_) is also considered as a green oxidant. The authors of [10], oxidized cyclohexanone to AA using H_2_O_2_ in the presence of heterogeneous iron oxides incorporated mesoporous carbon, resulting 93% CH=O conversion and 87% AA selectivity. Equally high AA yields, over 70%, were obtained when Mn(II) complexes were employed in CH=O oxidation by H_2_O_2_ [11]. However, H_2_O_2_ method has limitations, such as high price and instability of the oxidant as well as the presence of water introduced to the system with oxidant.

Reports have also examined ozone for AA preparation. For example, direct AA synthesis from CH was achieved using ozone and Fe(II) scorpionate complex and pyrazine carboxylic acid as an additive, generating AA in 96% yield and 98% selectivity [12]. However, methods using ozone are costly and hazardous.

Alternative petrochemical methods are based on butadiene, hexane, and cyclohexene [2,13,14]. However, nowadays, the availability of such raw materials is limited. The utilization of biomass has much more prospective [15].

Herein, an alternative method for AA synthesis of industrial importance has been reported. The proposed novel method involves the one-step, solvent-free oxidation of a cyclohexane/cyclohexanone mixture, with a cheap oxidizing agent such as O_2_ or air under mild conditions in the presence of NHPI and transition metals as catalysts. NHPI is known as a highly active and abundant organocatalyst for free radical oxidation processes [16]; it is often used in combination with Co(II) or Mn(II) compounds as co-catalyst in polar solvents [17]. Reports have described NHPI activity towards the oxidation of CH or CH=O separately with O_2_ in AcOH, however, to best of our knowledge, the solvent-free CH/CH=O mixture oxidation has yet to be examined. It was hypothesized that the presence of polar CH=O ensured the homogeneity of the reaction mixture, hence, the addition of a polar solvent was not necessary to solubilize NHPI. It is known that the addition of ketones as an initiator positively effects the oxidation of hydrocarbons with O_2_, including the addition of cyclohexanone or butanone to CH [18].

## 2. Materials and Methods

### 2.1. Materials

Cyclohexanone (Grupa Azoty S.A., Puławy, Poland, ≥99.9%), cyclohexane (Grupa Azoty S.A., Puławy, Poland, ≥99,9%), cyclohexanol (Grupa Azoty S.A., Puławy, Poland, ≥99.9%), manganese(II) acetylacetonate (Sigma-Aldrich, 99%), cobalt(II) acetylacetonate (Sigma-Aldrich, 97%), sulfuric acid (Chempur, 95%), methanol (Chempur, 99.8%), acetic acid (Chempur, 99.5%), acetonitrile (Chempur, 99.5%), benzonitrile (Chempur 99.5%), *N*-hydroxyphthalimide (Sigma-Aldrich, 97%).

### 2.2. Oxidation Process

The study was performed in a pressure reactor (Autoclave Engineers Inc., Erie, PA, USA) made of Hastelloy C-276 steel with a capacity of 100 mL.

The appropriate amount of raw materials and additives were introduced into the reactor. The reactor was then purged twice with oxygen or air, stirred at 200 rpm, and heated to the specified temperature. Then, the oxidizing agent was introduced into the reactor under the specified pressure (1–2 MPa) and the stirring rate was increased to 1000 rpm. The oxidation reaction was carried out without the flow of the oxidizing agent for 2–8 h after reaching the set temperature. When the pressure decreased during the reaction, it was supplemented through the introduction of oxygen. The composition of obtained products was determined by GC analysis.

### 2.3. GC Analysis

The composition of the post-reaction mixture was determined by the internal standard method using an Agilent Technologies 7890C gas chromatograph equipped with a FID detector, autosampler, ZB-5HT column (30 m × 0.25 mm × 0.25 µm) and helium as a carrier gas.

Detailed information on the reactor equipment, process procedures and GC analysis are provided in Appendix A.

## 3. Results and Discussion

This study examines the oxidation of CH/CH=O mixture to AA with O_2_ or air under pressure, without solvent, using NHPI, and Co(acac)_2_ and Mn(acac)_2_ as catalysts. The composition of obtained products were determined by means of GC analysis; i.e., amount of unreacted raw materials (CH, CH=O), the main product (AA), major by-products such as glutaric acid (GA) and succinic acid (SA), and intermediate product cyclohexanol (CH-OL) (Figure 2).

### 3.1. The Influence of Solvent

First, the conversion of the raw materials and selectivity to AA and to the main by-products obtained in the oxidation of CH/CH=O mixture (1:1 v/v) without solvent and in acetonitrile (MeCN), benzonitrile (PhCN) or AcOH were compared (Table 1). The reactions were carried out at 100 °C, under 1 MPa pressure.

It was found that under the applied conditions, neat CH did not oxidize (Entry 1). When CH was oxidized in the mixture with CH=O, its oxidation rate significantly increased (Entry 2). After 2 h, 37% conversions of CH was achieved. This was due to the increase in polarity of the reaction mixture, which improved NHPI solubility. Additionally, according to the literature, even small amounts of CH=O can accelerate CH oxidation [19]. Concurrently, 68% conversion of CH=O was obtained. For comparison, CH oxidation in AcOH as solvent was also carried out under the same conditions (Entry 6). CH conversion and AA selectivity of 39 and 31% were achieved, respectively.

The probable mechanism of oxidation of CH/CH=O mixture is presented in Figure 3.

Cobalt and manganese compounds as well as peroxide radicals can successfully generate the active PINO radical from NHPI. CH=O is known to undergo oxidation much easier than CH, due to the lower activation energy of the hydrogen atom abstraction reaction from the carbon atom at the β position [19]. Therefore, it is assumed that first PINO abstracts hydrogen from CH=O, then the alkyl radical (1) reacts which O_2_ molecule and the respective peroxy radical is formed (2) [20]. The peroxy radical (2) along with PINO can initiate the CH oxidation by the abstraction of hydrogen from the CH molecule. The obtained cyclohexyl hydroperoxide decomposes in the presence of cobalt or manganese compounds to cyclohexanol or cyclohexanone. The subsequent reactions leading to AA are also presented in Figure 3. Herein, it has been found that at higher temperature, the share of SA and GA formation reactions from the radical (3) is higher.

When CH/CH=O mixture oxidation was carried out in polar solvents (AcOH, MeCN, PhCN) higher CH conversions, ranging from 53% to 68% (Entries 3–5) was obtained. The highest selectivity to AA (49%) was obtained using AcOH. Nevertheless, it is extremely important from an industrial and environmental perspective to eliminate corrosive volatile organic solvents, allowing the oxidation processes to be carried out in equipment made from cheaper materials, which may lead to reduction of the costs of construction. Moreover, the lack of solvent may increase the product yield per reactor unit volume and eliminating the costs of solvent handling. Therefore, despite lower AA selectivity, further studies were carried out in solvent-free conditions. The investigation studied the influence of the following parameters on the composition of CH/CH=O oxidation products:type and amount of catalyst,cyclohexane to cyclohexanone ratio (*v*/*v*),temperature,time,type of oxidizing agent.

### 3.2. Influence of Catalyst Type and Amount

The effect of the amount of Co(acac)_2_, Mn(acac)_2_ and NHPI on the oxidation reaction of CH/CH=O mixture was examined at 100 °C using O_2_ at 1 MPa (Table 2).

The results showed that when the amount of Co(acac)_2_ increased from 0.05 to 0.25 mol% (Entries 1–3 and 5–7), an increase in CH conversion was observed, however, an undesirable decrease in AA selectivity also occurred. When the amount of Mn(acac)_2_ was increased to 0.125 mol% (Entries 9–11), CH=O conversion and AA selectivity increased, but CH conversion only slightly changed. Therefore, the obtained results indicate that the presence of both metal compounds is essential.

Addition of ≥0.5 mol% NHPI increased CH conversion from 41 to 51%, but did not increase AA selectivity (Entries 13–16). At 2.5 mol% NHPI, CH=O conversion decreased. This may be due to NHPI not completely dissolving in the system, which negatively influenced the free radical oxidation process with O_2_, as previously reported [21].

### 3.3. Effect of CH/CH=O Ratio

The studied oxidation reaction was conducted using various ratios of CH/CH=O at 100 °C with O_2_ at 1 MPa. The obtained conversion and selectivity results are shown in Table 3. For comparison, undiluted CH and CH=O were oxidized separately under the same conditions.

In the control reactions, CH oxidation did not occur (Entry 1), whereas CH=O conversion was 55% and AA selectivity was 30% (Entry 2). When, CH/CH=O mixtures were used CH and CH=O conversions as well as AA selectivity increased. AA was obtained with the highest selectivity of 46% after the oxidation of 3:1 CH/CH=O mixture (Entry 4). The product obtained by oxidation of 1:1 CH/CH=O mixture is presented in Figure 1 as an example.

The observed influence of CH/CH=O ratio was probably related to the change in the polarity of the mixture, which affected the course of the parallel and subsequent reactions taking place in the system.

The reaction mixtures obtained after 30 min, 1 and 2 h of reaction are shown in Figure 1. At the beginning of the reaction, the components of Co/Mn/NHPI catalytic system dissolve in the CH/CH=O mixture. However, due to the decrease in CH=O concentration with the reaction progress, catalysts precipitate out along with the formed acids (AA, GA and SA) coloring them. 

The advantage of a solvent-free process is the simple separation of crude AA, GA, and SA by post-reaction mixture filtration. It has been shown that almost the whole AA, GA and SA precipitated out from the reaction mixture. The crude mixture of acids obtained must be purified by crystallization from polar solvents, which allows to obtain AA with commercial purity.

### 3.4. Influence of Temperature and Reaction Time

The parameters that significantly impacted product composition were temperature and time. Table 4 shows their influence on the CH/CH=O mixture oxidation reaction conducted in the range of 70 to 100 °C for 2–8 h (Table 4).

As the temperature increased from 70 to 90 °C, both CH and CH=O conversion (Entries 1–3) increased, but not AA selectivity. At 100 °C (Entry 4) a further decrease in AA selectivity was observed but also the conversion of raw materials. Additionally, reduced CH=O conversion highlighted that NHPI deactivation occurred under applied conditions. Relatively high AA selectivity (48–49%) was obtained between 80 and 90 °C.

In the initial stage of the reaction (up to 2 h), CH=O was oxidized approx. twice faster than CH. Further reaction time extending to 8 h resulted in increased CH and CH=O conversion, but AA selectivity decreased (Entries 5–8). The practically constant selectivity to GA and SA indicates an increase of share of reactions to other products

### 3.5. Influence of the Oxidizing Agent

Implementation of the process on an industrial scale requires the use of a cheaper and safer oxidizing agent, such as air. Table 5 shows the conversion and selectivity of the main products obtained in reaction using oxygen and air.

Unfortunately, in the presence of air and at 2 MPa pressure, the expected reaction did not occur and the rate of CH and CH=O oxidation was significantly lower.

## 4. Conclusions

The study of solvent-free oxidation of CH/CH=O with O_2_, directly synthesized AA in the presence of Co(acac)_2_, Mn(acac)_2_ and NHPI catalytic system was reported.

The obtained results showed that CH/CH=O mixture under applied mild conditions oxidized faster than CH and CH=O separately. This was due to the greater oxidizability of CH=O compared to CH. The peroxyl radicals formed by CH=O oxidation initiated the oxidation of the less reactive CH. Additionally, CH=O increased the polarity of the reaction mixture, promoting the solubility of NHPI. As a result, the utilization of a polar solvent was eliminated, which is important for industrial processes, both from a technological, economic and environmental point of view.

The highest AA selectivity and conversion in the solvent-free oxidation of CH/CH=O mixture were obtained using Mn(acac)_2_ 0.25 mol%, Co(acac)_2_ 0.05 mol% and NHPI 2.5 mol% at 90 °C for 2 h.

## Data Availability

All data is available in the manuscript or upon request to the corresponding author.

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
