# Peer review of "Oxidation of Cyclohexane/Cyclohexanone Mixture with Oxygen as Alternative Method of Adipic Acid Synthesis"

_materials, 2022, doi:10.3390/ma16010298_

Round 1
Reviewer 1 Report
Comments
Author has presented an article on “Oxidation of cyclohexane / cyclohexanone mixture with oxygen as alternative method of adipic acid synthesis” The manuscript in the current state cannot be accepted for publication. The article is well documented and has reasonable, logical and scientific analysis.
1. The authors should incorporate the role of the catalyst in the proposed reaction mechanism.
2. The authors should also describe how the catalytic system helps in the reaction process in a dedicated paragraph in the manuscript.
3. The authors have used Co and Mn transition metals in combination with NHPI as catalyst. What about combinations with other transition metals such as V, Cu or Fe? Have the authors considered bi-component transition metal-based heterogeneous catalytic systems?
4. The manuscript has many grammatical errors which need to be rectified.
Thus, the manuscript requires some revision can be accepted after incorporating the above suggestions.
Author Response
Thank you for the review. We have amended the paper taking into account valuable comments of Reviewers.
- The authors should incorporate the role of the catalyst in the proposed reaction mechanism.
The role of cobalt and manganese ions in the proposed mechanism has been shown in Scheme 3.
- The authors should also describe how the catalytic system helps in the reaction process in a dedicated paragraph in the manuscript.
The description of the reaction’s mechanism was supplemented as follows:
"Cobalt and manganese compounds as well as peroxide radicals can successfully generate the active PINO radical from NHPI. CH=O is known to undergo oxidation much easier than CH, due to the lower activation energy of the hydrogen atom abstraction reaction from the carbon atom at the β position [19]. Therefore, it is assumed that first PINO abstracts hydrogen from CH=O, then the alkyl radical (1) reacts which O2 molecule and the respective peroxy radical is formed (2) [20]. The peroxy radical (2) along with PINO can initiate the CH oxidation by the abstraction of hydrogen from the CH molecule. The obtained cyclohexyl hydroperoxide decomposes in the presence of cobalt or manganese compounds to cyclohexanol or cyclohexanone. The subsequent reactions leading to AA are also presented in Scheme 3. Herein, it has been found that at higher temperature, the share of SA and SB formation reactions from the radical (3) is higher."
- The authors have used Co and Mn transition metals in combination with NHPI as catalyst. What about combinations with other transition metals such as V, Cu or Fe? Have the authors considered bi-component transition metal-based heterogeneous catalytic systems?
We investigated the influence of various catalytic systems containing transition metals compounds such as Co, Mn, Fe, V on the oxidation of cyclohexane, cyclohexanone and their mixtures.
- The Co/Fe or Co/V systems are mainly used in the industrial process of cyclohexane oxidation to cyclohexanone and cyclohexanol. The presence of Co compounds ensures higher cyclohexane conversions. The addition of Fe or V enhances the decomposition of cyclohexane hydroperoxide to cyclohexanone, which is more advantageous from a technological point of view. However, low yields of adipic acid are obtained in the presence of these both systems.
- In the presence of Co/Mn system higher yields of adipic acid are achieved by oxidation of cyclohexane, cyclohexanone or their mixture. However, to achieve the required reaction rate temperature above 120 ºC is needed, which unfortunately results in low AA selectivity.
- The addition of NHPI to the Co/Mn system allows the oxidation reaction to be carried out at lower temperature and to obtain AA with higher selectivity.
- The heterogeneous catalysis is also the subject of our studies. Nevertheless, the use of solid catalyst in studied process requires addition of solvent to dissolve formed AA.
- The manuscript has many grammatical errors which need to be rectified.
Manuscript has been proofread by a native speaker of English
Reviewer 2 Report
The MS of D. Lisicki, B. OrliÅ„ska, A.A. Marek, J. BiÅ„czak, K. Dziuba and T. Martyniuk (materials-1999655), entitled “Oxidation of cyclohexane / cyclohexanone mixture with oxygen as alternative method of adipic acid synthesis”, describes the elaboration of solvent-free procedure of oxidation of cyclohexane-cyclohexanone mixture to a mixture of dicarboxylic acids with adipic acid as its main component.
The main aim of this MS is developing of the “greener“ method for industrial synthesis of adipic acid, which is a multi-tonnage starting compound for polyamide polymers production. It seems, the Authors achieve quite good results, thus this paper can be published in Materials.
However, the presentation of the Authors’ results could be improved:
1) All the abbreviations (AA, CH, CH-ON, NHPI, etc.) should be decrypted at their first appearance, let just even in the Abstract (and probably, in Keywords section as well?). What do you think, would CH=O be better abbreviation for cyclohexanone (for readers understanding), than CH-ON? BA seems common for Polish people (bursztyn, kwas bursztynowy), but not for English speaking persons. Probably, SA is a better abbreviation for succinic acid?
2) Page 4, Table 1, Entry 1. Zero conversion is not a surprise – all the listed additives are (practically) insoluble in cyclohexane. I’d like to recommend adding to this table results for a mixture of cyclohexane and one of the listed solvents (acetic acid, for example) containing no CH-ON. Thus, the effect of cyclohexanone will be emphasized more earnestly.
3) Page 7, lines 208/214. The Authors should comment three photos in Figure 1. What do they mean: starting moment, reaction in progress, reaction is completed? Or something else? If the first photo really shows the initial reaction mixture, it indicates bad solubility of the additives even in CH/CH-ON mixture, am I right? Third photo demonstrates the polluted (precipitated) target product – a mixture of three dicarboxylic acids, each of them is a colorless solid. Were there any intermediate cleaning procedures before the GC analyses? Or esterification plays this role?
4) GC analyses of AA, GA, BA mixture via intermediate esterification (Supplementary section). The Authors claim total >98.5% conversion yield. Did the Authors check the yields of this reaction (at reported conditions) separately for each of these three acids? If they are numerically not equal one to another, the applied analytical procedure would not work properly.
5) Page 8, line 235, Table 5. Do O2 content in pure oxygen at 1MPa and in air at 2 MPa is equivalent? No wonder, that using air gave worse results compared to oxygen.
6) Do the Authors validate their analytical procedures by the method “entered-found”? What is the mean error of the numerical data presented in Tables 1-5? If such results are available, please include them at least into Supplementary section.
7) The Authors declare the general need for creation of economically attractive methods for industrial production of adipic acid (page 2, line 49). This is not necessary for this paper, but if the Authors can present costs comparison of their procedure with traditional CH/CHOH/CHON oxidation by HNO3, this will significantly improve the quality of the presented material and underline its importance to chemical industry.
Author Response
Thank you for the review. We have amended the paper taking into account valuable comments of Reviewers.
- All the abbreviations (AA, CH, CH-ON, NHPI, etc.) should be decrypted at their first appearance, let just even in the Abstract (and probably, in Keywords section as well?). What do you think, would CH=O be better abbreviation for cyclohexanone (for readers understanding), than CH-ON? BA seems common for Polish people (bursztyn, kwas bursztynowy), but not for English speaking persons. Probably, SA is a better abbreviation for succinic acid?
All abbreviations have been corrected according to Reviewer’s suggestion.
- Page 4, Table 1, Entry 1. Zero conversion is not a surprise – all the listed additives are (practically) insoluble in cyclohexane. I’d like to recommend adding to this table results for a mixture of cyclohexane and one of the listed solvents (acetic acid, for example) containing no CH-ON. Thus, the effect of cyclohexanone will be emphasized more earnestly.
The table 1 is completed with example 6 relating to the oxidation of CH in AcOH. CH conversion of 39% and AA selectivity of 31% were obtained. The comments has also been added as follows:
“For comparison, CH oxidation in AcOH as solvent was also carried out under the same conditions (Entry 6). CH conversion and AA selectivity of 39 and 31% were achieved, respectively."
- Page 7, lines 208/214. The Authors should comment three photos in Figure 1. What do they mean: starting moment, reaction in progress, reaction is completed? Or something else? If the first photo really shows the initial reaction mixture, it indicates bad solubility of the additives even in CH/CH-ON mixture, am I right? Third photo demonstrates the polluted (precipitated) target product – a mixture of three dicarboxylic acids, each of them is a colorless solid. Were there any intermediate cleaning procedures before the GC analyses? Or esterification plays this role?
The publication was supplemented with the following text:
“ The reaction mixtures obtained after 30 minutes, 1 and 2 h of reaction are shown in Figure 1. At the beginning of the reaction, the components of Co / Mn / NHPI catalytic system dissolve in the CH / CH = O mixture. However, due to the decrease in CH = O concentration with the reaction progress, catalysts precipitate out along with the formed acids (AA, GA and SA) coloring them.
The advantage of a solvent-free process is the simple separation of crude AA, GA, and SA by post-reaction mixture filtration. It has been shown that almost the whole AA, GA and SA precipitated out from the reaction mixture. The crude mixture of acids obtained must be purified by crystallization from polar solvents, which allows to obtain AA with commercial purity. "
The purification of crude mixture of obtained acids is also a subject of our studies, but cannot be published due to the industrial cooperation.
In our research, methanol esterification of AA, GA and SA was performed only for the reason of quantitative analysis by means of gas chromatography (GC). Low volatility of these acids limits direct GC analysis. The esterification to methyl diesters allows to increase volatility and perform reliable chromatographic analyzes. It is possible to perform analyzes without esterification using liquid chromatography (HPLC). However the GC technique is faster and better.
- GC analyses of AA, GA, BA mixture via intermediate esterification (Supplementary section). The Authors claim total >98.5% conversion yield. Did the Authors check the yields of this reaction (at reported conditions) separately for each of these three acids? If they are numerically not equal one to another, the applied analytical procedure would not work properly.
Gas Chromatography analysis (GC) is a common method for determining dicarboxylic acids after prior esterification with methanol. The aliphatic dicarboxylic acids are easily esterified with methanol at ambient temperature in a short time in the presence of an acid catalyst.
We elaborated the esterification method of all linear aliphatic dicarboxylic acids (C4-C20). The method we use enables us to obtain an acid conversion of over 98.5% for any concentration of dicarboxylic acids. We confirmed this by GC, GC MS HPLC and acid number determination.
For example, if we assume that 1 g of the sample contains only adipic acid (concentration 100%), then after introducing 12 ml of methanol 21.7 - fold molar excess of methanol was used, which allows the reaction equilibrium to be shifted towards dimethyl esters. GC / MS analyzes confirmed the presence of only traces of monoester and no dicarboxylic acids were detected by HPLC. Nevertheless, in the samples the concentrations of AA, GA and SA are significantly lower than 100%, so the excess of methanol is significantly greater than 21.7 times, which makes it possible to obtain diesters with even higher conversion of dicarboxylic acid.
The composition of oxidation products for selected samples was also confirmed by GC after prior silylation of acids with 1,1,1,3,3,3-hexamethyldisilazane. The silylation was performed in pyridine in the presence of trifluoroacetic acid with the same results.
- Page 8, line 235, Table 5. Do O2content in pure oxygen at 1MPa and in air at 2 MPa is equivalent? No wonder, that using air gave worse results compared to oxygen.
The content of O2 in air atmosphere at 2 MPa is much less than under 1 MPa of pure oxygen. To perform process with the same content of oxygen, the pressure of air should be about 5 MPa. Unfortunately, we do not have a pressure reactor supplied with required equipment to perform reactions under such conditions.
- Do the Authors validate their analytical procedures by the method “entered-found”? What is the mean error of the numerical data presented in Tables 1-5? If such results are available, please include them at least into Supplementary section.
The composition of the reaction products was determined by the method of the calibration curve with an internal standard. For each tested substance, a standard curve was prepared based on 7 solutions with different concentrations. Each reference sample was analyzed 3 times. The graph of the dependence y = ax + b was obtained where: y = (surface area of the tested substance / surface area of the standard (toluene), x = (mass of the tested substance / mass of standard (toluene). The coefficients "a" and "b" were determined successively. The curve makes it possible to determine the mass of the substance which was analyzed. Each of the samples was two times determined using GC.
The error of the chromatographic analysis was found to be 2%.
- The Authors declare the general need for creation of economically attractive methods for industrial production of adipic acid (page 2, line 49). This is not necessary for this paper, but if the Authors can present costs comparison of their procedure with traditional CH/CHOH/CHON oxidation by HNO3, this will significantly improve the quality of the presented material and underline its importance to chemical industry.
The development of a new method of producing adipic acid is extremely important for ecological and economic reasons. The production costs of AN are related to the costs of greenhouse gas emissions. Market studies suggest that changes to the EU Directives are possible, which will prohibit the use of HNO3 as an oxidizing agent in industrial processes for the production of chemical compounds, including monomers for further syntheses.
The economic analysis of obtaining AA from CH and CH=O is one of the most important milestones of our research project , but it cannot be published.
Round 2
Reviewer 1 Report
Authors have incorporated all the comments and thus the manuscript can be accepted in current form.